# Function-based high-throughput screening for antibody antagonists and agonists against G protein-coupled receptors

Huanhuan Ren[1,2], Jian Li[1], Ning Zhang[1], Liaoyuan A. Hu [1], Yingli Ma [1], Philip Tagari[3], Jianqing Xu [2✉] & Mei-Yun Zhang [1✉]

Hybridoma and phage display are two powerful technologies for isolating target-specific monoclonal antibodies based on the binding. However, for complex membrane proteins, such as G protein-coupled receptors (GPCRs), binding-based screening rarely results in functional antibodies. Here we describe a function-based high-throughput screening method for quickly identifying antibody antagonists and agonists against GPCRs by combining glycosylphosphatidylinositol-anchored antibody cell display with β-arrestin recruitment-based cell sorting and screening. This method links antibody genotype with phenotype and is applicable to all GPCR targets. We validated this method by identifying a panel of antibody antagonists and an antibody agonist to the human apelin receptor from an immune antibody repertoire. In contrast, we obtained only neutral binders and antibody antagonists from the same repertoire by phage display, suggesting that the new approach described here is more efficient than traditional methods in isolating functional antibodies. This new method may create a new paradigm in antibody drug discovery.

[1] Amgen Research, Amgen Asia R&D Center, Amgen Biopharmaceutical R&D (Shanghai) Co., Ltd., 13th Floor, Building No. 2, 4560 Jinke Road, Zhangjiang, Shanghai 201210, China. [2] Shanghai Public Health Clinical Center, Institutes of Biomedical Sciences, Fudan University, PI's Office IV, Scientific Research Center Building, 2901 Caolang Road, Jin Shan District, Shanghai 201508, P. R. China. [3] Amgen Research, Therapeutic Discovery, Amgen Inc., One Amgen Center Dr, Thousand Oaks, CA 91320, USA. ✉email: meiyunz@amgen.com; xujianqing@shphc.org.cn

The history of antibody drug discovery reflects the advancement of antibody screening technology. The invention of hybridoma technology in 1975 led to the first paradigm in antibody drug discovery[1]. The first monoclonal antibody (mAb), muromonab-CD3 (OKT3), approved by FDA in 1986 was derived from hybridoma screening[2]. The invention of recombinant DNA technology in the early 1970's allowed gene cloning and subsequently protein engineering for desired properties[3–7]. The invention of polymerase chain reaction (PCR) in 1983 and phage display technology in 1985 further enabled batch amplification of antibody genes and isolation of target-specific binding antibodies from combinatorial antibody libraries in a high-throughput manner, which created a paradigm shift in antibody drug discovery[8,9]. In 2002, adalimumab became the first phage display-derived antibody approved by FDA[10]. It was also the first approved human mAb and is currently the best-selling antibody drug on the market[11]. Phage display is now one of the most widely used technologies for antibody discovery and engineering[12]. Following the invention of phage display technology, various other display technologies, such as yeast display[13] and ribosome display[14], emerged, which further enhanced rapid discovery of antibodies. Compared to hybridoma technology that retains the natural paring of the heavy and light chains, display technologies allow random pairing of the heavy and light chains, which generates additional antibody diversity. Display technologies also facilitate generating antibodies against low immunogenic targets and engineering antibodies for desired properties. However, hybridoma screening and these display technologies are generally based on antibody binding to recombinant antigens. Following hybridoma screening and phage (or yeast, or ribosome) display, expression, and purification of isolated clones and characterization of purified antibodies for biofunction are mandatory. The workload for antibody production and characterization of isolated clones could be enormous, but the chance to obtain functional antibodies (antibody antagonists and agonists) may be slim.

G protein-coupled receptors (GPCRs) play important roles in many (patho)physiological processes and represent the largest family of drug targets. However, the conformation flexibility and low immunogenicity and antigenicity of GPCRs pose challenges to generating target-specific antibody binders, let alone functional antibodies against GPCRs. If the immunization was successful, hybridoma screening or phage-displayed library panning and screening may lead to the identification of neutral binders and, sometimes, antibody antagonists[15,16], but rarely antibody agonists. GPCR activation is a rather complex process. It involves a series of sequential conformational changes that are initiated by ligand engagement and coordinated by the transmembrane domains to eventually expose the G protein binding site in the intracellular side of the receptor. One-third of approved drugs target GPCRs[17,18], among which, only two are mAbs, Erenumab, and Mogamulizumab, and both were approved in 2018[19,20]. Erenumab is a Calcitonin gene-related peptide (CGRP) receptor antibody antagonist derived from XenoMouse immunization and hybridoma screening[21]. Mogamulizumab is a humanized, defucosylated CCR4-binding mAb derived from mouse immunization and hybridoma screening. Mogamulizumab is a neutral binder and functions only in vivo through antibody-dependent cellular cytotoxicity for the treatment of adult T-cell leukemia-lymphoma[22]. Ten GPCR-targeting mAbs are currently in clinical trials[19], among which, GMA102 is a humanized GLP-1R-binding mAb fused to GLP-1 peptide, and the rest are antibody antagonists or neutral binders conjugated to isotopes. None of these antibodies are antibody agonists. Due to the challenges to identifying antibody agonists to membrane receptors, it has been a common practice to use natural peptide ligands or ligand

mimetics fused to Fc or other half-life extending moieties to agonize the receptors for the treatment of diseases. This approach may have the risk that the endogenous peptide ligand is wiped out by anti-drug neutralizing antibodies that may be induced by repeated administration of peptide ligand mimetics[23]. Compared to natural peptide ligands and ligand mimetics, antibody agonist is preferred for its predictable pharmacokinetics and well-developed manufacturing process. Therefore, developing a function-based antibody high-throughput screening method would be of great value for identifying mAbs with desired biofunction to complex membrane protein targets.

We aim to develop a universal high-throughput screening method for direct identification of antibody antagonists and agonists to any GPCR targets of interest. We took advantage of glycosylphosphatidylinositol (GPI) anchoring system and β-arrestin recruitment-based reporter assay. GPI allows proteins, including antibodies, to anchor on the cell surface and keep the anchored proteins functional[24–26]. More importantly, GPI anchoring sequences can be recognized by GPI transamidase for the attachment to the lipid rafts[27]. GPCRs also preferentially diffuse to the lipid rafts for trafficking and signaling[28–30]. The co-localization of GPCR targets and GPI-anchored antibodies in the lipid rafts would facilitate the interaction between the target and the anchored antibody on the same cell surface. β-arrestin recruitment upon ligand stimulation is a common phenomenon among GPCRs regardless of G protein-mediated downstream signaling pathways (Gs, or Gi, or $G_{12/13}$, or Gq)[31]. Therefore, GPI-anchored antibody library cell display in combination with β-arrestin recruitment-based reporter assay may be applied to isolating functional antibodies for any GPCR targets.

We validated this new method using human apelin receptor (APJ), a Gαi-coupled class A GPCR, as a model GPCR target. APJ is involved in cardiovascular function and agonizing APJ is beneficial for the treatment of chronic heart failure[32–34]. APJ is also involved in angiogenesis and elevation of APJ level was observed in tumors[35–38]. Therefore, APJ antibody antagonists may have the potential for the treatment of APJ-overexpressing tumors by inhibiting tumor angiogenesis. We previously immunized a camel with APJ nanodiscs and constructed an immune camelid single domain antibody (sdAb) library in a phagemid vector. From the panning and screening of phage-displayed sdAb library against APJ liposomes, we obtained a panel of potent sdAb antagonists and a panel of neutral binders. No sdAb agonist was identified by the phage display effort[39]. To validate the new method described in this study and to identify APJ antibody antagonists and agonists, we transfer the APJ immune sdAb library to a lentiviral vector containing decay accelerating factor (DAF) sequence for anchoring antibodies to GPI. We then transduce the recombinant lentivirus library into U2OS-derived Tango β-arrestin reporter cells stably expressing human APJ. Following β-arrestin recruitment-based cell sorting and screening, we identify a panel of sdAb antagonists with diverse epitope coverage and, for the first time to our knowledge, a potent sdAb agonist to human APJ.

## Results

**Proof of concept study and generation of a GPI-anchored sdAb library.** Lentivirus transduction is widely used for generating stable cell lines. We constructed a lentiviral expression vector, designated pHBLV-puro-GPI, which contains a CMV promoter, a multiple cloning site (MCS), a his6 tag, a $(G_4S)_5$ linker and DAF sequence (Supplementary Fig. 1a, b). Covalently linked DAF allows his-tagged sdAbs to anchor to GPI through a flexible linker (Supplementary Fig. 1b). For a proof of concept (POC) study, we subcloned four sdAb genes to the pHBLV-puro-GPI vector,

including APJ neutral binding sdAb JN126, agonistic antibody-ligand fusion JN126P13, antagonistic sdAb JN241, and a control sdAb E3 recognizing an irrelevant GPCR. JN126 and JN241 are APJ sdAbs previously isolated from the APJ immune library by phage display, while JN126P13 is an antibody-ligand fusion with apelin 13 covalently linked to the C-terminus of the neutral binding sdAb JN126. Following the generation of recombinant lentiviruses and the transduction of APJ stably expressing CHO-k1-derived PathHunter β-arrestin reporter cells, we measured antibody surface expression level by flow cytometry and tested the transduced cells for biological function by β-arrestin recruitment assay. The result from flow cytometry showed that all four tested sdAbs were highly expressed and well displayed on the cell surface. Treatment of these cells with phosphatidylinositol phospholipase C (PI-PLC) substantially reduced the surface density of these antibodies, confirming that they were indeed anchored to GPI (Supplementary Fig. 1c). In PathHunter β-arrestin assay, GPI-anchored sdAb JN241 and antibody-ligand fusion JN126P13 were confirmed to be antagonist and agonist, respectively, while neutral binding sdAb JN126 and control sdAb E3 were non-functional (Supplementary Fig. 1d, e). We further tested their corresponding Fc fusions for GPI-anchored cell display. The results showed that sdAbs JN126 and JN241 fused to human Fc, JN126Fc and JN241Fc, and JN126Fc fused to apelin13, JN126FcP13, also well displayed on cell surface through GPI and kept their respective functional characteristics (Supplementary Fig. 1f–h).

PathHunter β-arrestin reporter assay system is suitable for monoclonal cell screening, but not suitable for sorting surface-displayed antibody cell library based on function because neither the fluorescent substrate for the reporter enzyme nor the product resides in the cells after the cells are washed. Therefore, we switched to the Tango β-arrestin assay system[40,41], in which both the fluorescent substrate and the product retain within the cells before and after hydrolysis by the β-lactamase reporter enzyme. We repeated the POC study with Tango/APJ β-arrestin reporter cells, in which APJ gene is stably integrated into the genome, as well as a transcription factor linked to the C-terminus of APJ through a tobacco etch virus (TEV) protease site. This reporter cell line also stably expresses a β-arrestin and TEV protease fusion protein. Upon ligand binding and receptor activation, β-arrestin-TEV fusion proteins are recruited to the intracellular side of the receptor, leading to the cleavage of transcription factor and subsequent expression of β-lactamase reporter gene that can be monitored by β-lactamase-mediated enzymatic reactions (Supplementary Fig. 2a). Results from the functional assay in Tango/APJ β-arrestin reporter cells confirmed that all four tested sdAbs behave the same as that in APJ PathHunter reporter cells (Fig. 1a and Supplementary data). Dot plots of JN241-expressing cells showed a clear gating window for the cells with antagonistic antibodies anchored on the cell surface (Fig. 1b and Supplementary data). Similarly, dot plots of JN126P13-expressing cells suggested a possible gating window for the cells with agonistic antibodies anchored on the cell surface (Fig. 1c and Supplementary data). To enrich for APJ-specific sdAb clones, we did one round panning of the phage-displayed immune library against 293FT/huAPJ cells, then batch-transferred the inserts from the panned library to the lentiviral vector pHBLV-puro-GPI. We constructed a GPI-anchored sdAb library in pHBLV-puro-GPI with a size of $4 \times 10^7$ (10 fold larger than the panned phage library in size; Supplementary Fig. 2b). To bias the population toward a single sdAb per cell, we used a low multiplicity of infection of 0.3 in lentivirus transduction. Transduced cells account for 27% by flow cytometry 48 h post lentivirus infection. To eliminate non-transduced cells, 0.5 μg per ml puromycin was added to the cell-displayed sdAb library. Following the treatment

with puromycin for a week, we carried out flow cytometry to determine the positive cell percentage and it reached 80% by staining the cells with anti-his antibody (Supplementary Fig. 2c).

**Function-based cell sorting and screening for sdAb antagonists.** Since we knew that sdAb antagonists exist in the immune repertoire, we first validated the function-based cell sorting and screening for APJ sdAb antagonists. Prior to the cell sorting, GPI-anchored sdAb library ($1 \times 10^8$ cells, in total) was treated with 80% maximal effective concentration (EC$_{80}$) of apelin 13 followed by substrate loading. Cells with low product intensity at 460 nm and high substrate intensity at 530 nm were sorted out (Fig. 2a and Supplementary data). Sorted cells were cultured and subjected to the second round of cell sorting. After three rounds of cell sorting, cells with antagonist activity were enriched based on the dramatically decreased ratio of product/substrate intensity (Fig. 2b and Supplementary data). Single cells from the third round sorted library were plated into 96-well plates and a total of 127 single-cell clones were subsequently tested for antagonist activity by β-arrestin recruitment assay. In all, 104 out of 127 cell clones showed higher than 20% inhibition. Among them, 72 clones showed higher than 50% inhibition (Fig. 2c and Supplementary data). Antibody genes from 24 positive cell clones with higher than 50% inhibitory activity were rescued by reverse transcription PCR (RT-PCR) and sequenced. Ten unique clones were obtained and further characterized. Soluble sdAbs of all 10 clones showed antagonist activity in APJ PathHunter assay. Seven sdAbs, JN400, JN402, JN406, and JN411-414, had a half-maximal inhibition concentration (IC$_{50}$) below 200 nM. Two sdAbs, JN401 and JN409, showed an IC$_{50}$ over 200 nM (Fig. 2d and Supplementary data). One sdAb JN404 showed weak antagonist activity and the IC$_{50}$ was not determined. We did a correlation analysis between the percentage inhibition of the cell clones and the IC$_{50}$ of the nine sdAbs isolated from the corresponding cell clones. Notably, the percentage inhibition of a cell clone correlates well with the IC$_{50}$ of soluble sdAb isolated from the corresponding cell clone (Fig. 2e and Supplementary data).

**Sequence and epitope comparison of isolated APJ sdAb antagonists.** We further tested the isolated sdAb antagonists for antagonist activity by LANCE cAMP assay as previously described[42]. The IC$_{50}$s obtained by the cAMP assay correlate with those by the β-arrestin recruitment assay (Fig. 3a, b and Supplementary data). We next compared the ten sdAb antagonists isolated by function-based cell sorting and screening with the sdAb antagonists previously isolated by phage display from the same immune repertoire (Supplementary Fig. 3). Sequence comparison showed that one of the sdAb antagonists, JN414, shares the same sequence with JN241 previously isolated by phage display[39]. The other nine sdAbs differ from phage display-derived sdAb antagonists in amino acid sequence. We localized the epitopes of the newly identified 10 sdAb antagonists by flow cytometry with APJ E174A mutant [E174 is located on APJ extracellular loop (ECL) 2] and APJ domain-swap chimeras containing the N-terminus, or ECL1, or ECL3 from δ-Opioid Receptor (DOR) (Supplementary Fig. 4). Our previous study indicates that E174 is critical for the binding and function of sdAb antagonist JN241 and most of the other phage display-derived sdAb antagonists. Interestingly, we found that, unlike phage display-derived sdAb antagonists that bound dominantly to ECL2, the sdAb antagonists isolated by function-based cell sorting and screening bound to diverse epitopes on APJ which can be categorized into five groups (Fig. 3c and Supplementary data). The first three groups include all seven potent sdAb antagonists and their binding involves either ECL2 (group 1) or ECL1 (group 3), or

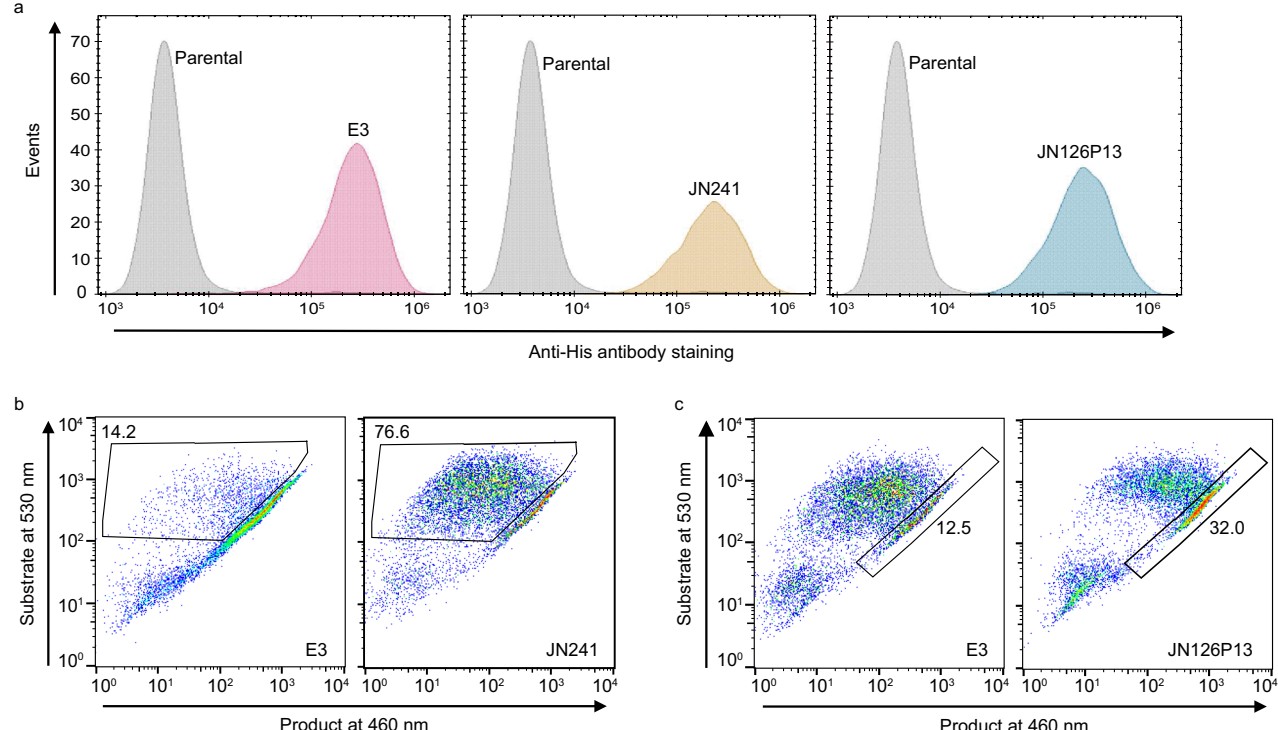

**Fig. 1 POC study in U2OS/APJ β-arrestin reporter cell line. a** Flow cytometry of Tango/APJ β-arrestin assay cells transduced with recombinant lentivirus containing the genes encoding sdAb E3, or JN241, or JN126P3. **b**, **c** Flow cytometry of Tango/APJ β-arrestin assay cells following β-arrestin recruitment assay at antagonist (**b**) or agonist (**c**) mode. The gated window and the percentage of gated cell population are shown. In the assay at antagonist mode, apelin 13 at $EC_{80}$ dose was added to the plates. Antibody antagonists negatively affect apelin 13 binding to the receptor, which causes decreased β-lactamase production and subsequent reduced substrate hydrolysis, leading to more substrate fluorescence (at 530 nm) and less product fluorescence (at 460 nm). In the assay at agonist mode, no ligand was added to the plates.

both (group 2). The remaining two groups with moderate to weak antagonist activities bound to the N-terminus and almost all three extracellular loops of APJ (group 4 and 5), indicating that they are more sensitive to the conformation integrity of the receptor (Fig. 3 and Supplementary data).

**Function-based cell sorting and screening for sdAb agonists**. It has rarely been successful in isolating antibody agonists to GPCRs by hybridoma technology and phage display. This could be ascribed to the rarity of antibody agonists in the repertoires and/or the inefficiency of the screening methods. Our previous phage display effort did not result in the isolation of APJ sdAb agonist from the camelid immune sdAb library. To investigate whether function-based cell sorting and screening of GPI-anchored sdAb library will increase the chance to identify sdAb agonists, we did β-arrestin recruitment-based cell sorting for the cells with high product intensity (signal at 460 nm) and low substrate intensity (signal at 530 nm) following the reporter substrate loading (Fig. 4a and Supplementary data). After three rounds of cell sorting, the percentage of cells with agonist activity increased. The results from the Tango assays showed increased agonist activity of the sorted cell population after each round of sorting, confirming the enrichment of cell clones with agonist activity (Fig. 4b and Supplementary data). Single cells from the 3rd round sorted library were grown out and a total of 72 cell clones were screened by Tango/APJ β-arrestin assay at agonist mode. Ten cell clones showed over 50% increase in agonist activity compared to the parental cells stimulated with 100 nM apelin 13. Nineteen cell clones showed 20–50% increase in agonist activity (Fig. 4c and Supplementary data). All agonistic cell clones were checked under confocal microscopy and the agonist activity was confirmed with

the presence of the product (in blue color) (Fig. 4d and Supplementary data). One sdAb agonist, designated JN300, was identified from one of the positive cell clones and extensively characterized.

We characterized soluble sdAb JN300 for binding and agonist activities (Fig. 5 and Supplementary data). JN300 specifically bound to human APJ-expressing cells (Fig. 5a and Supplementary data). We localized the epitope of JN300 by testing its binding to wild type (WT) APJ, APJ E174A mutant, and APJ/DOR loop-swap chimeras. We found that any changes to the N-terminus or to the three ECL loops substantially disrupted the binding of JN300 to APJ (Fig. 5b and Supplementary data), suggesting that the structural integrity of APJ receptor is critical for JN300-mediated receptor agonism. In both β-arrestin, cAMP, and PathHunter-based APJ internalizing assays, JN300 showed a half-maximal effective concentration ($EC_{50}$) of 80–90 nM (Fig. 5c–e and Supplementary data). We compared JN300 with the isolated ten sdAb antagonists in amino acid sequence and expression level (Supplementary Fig. 3 and Supplementary Table 1). We found that sdAb agonist JN300 did not show any special characteristics in V, D, and J gene usage and in CDR loop lengths except that the CDR1 of JN300 was short, but the yield of JN300 in 293F transient expression system was lower than that of the 10 sdAb antagonists. We are currently trying to co-crystalize APJ and JN300 complex. The resolution of the complex structure may shed light on JN300-mediated receptor agonism.

## Discussion

This study describes a new approach for efficient identification of antibody antagonists and agonists against GPCRs by combining GPI-anchored antibody cell surface display with β-arrestin

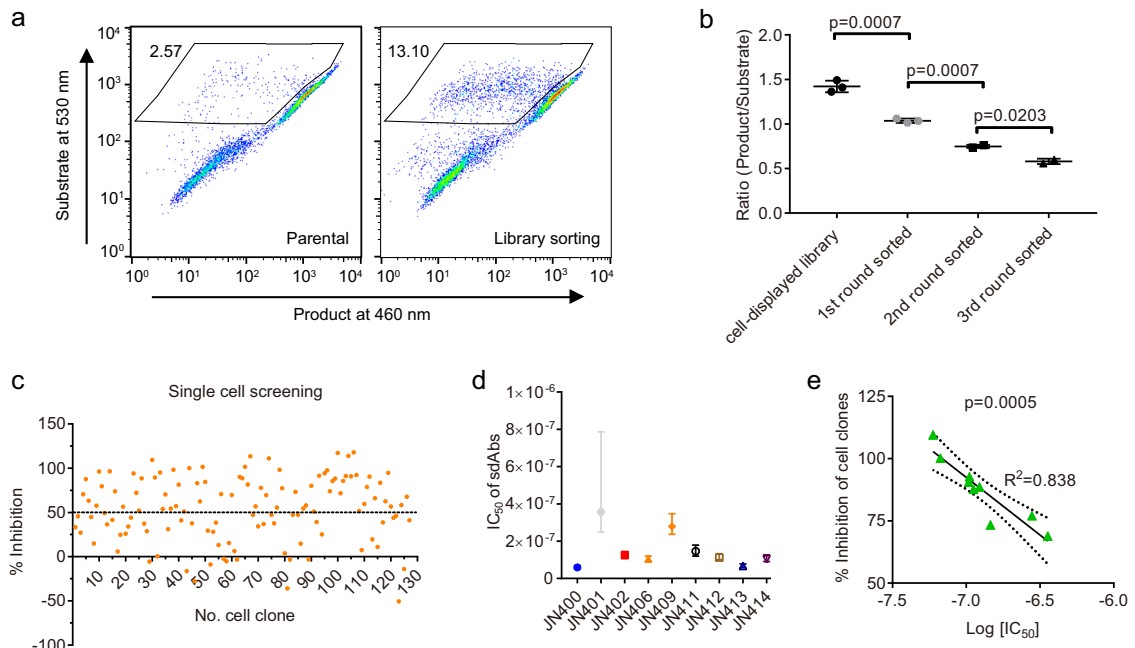

**Fig. 2 Function-based cell sorting and screening for sdAb antagonists. a** Function-based cell sorting for antagonist-expressing cell clones following the stimulation with apelin 13 for 12 h. The gated window and the percentage of gated cell population in the 3rd round cell sorting is shown. **b** Antagonist activity of each round sorted cell population in Tango/APJ β-arrestin assay. Data are expressed as mean with SD. Statistical analysis was done using unpaired two-tailed *t* test. The lower product/substrate ratio, the higher antagonist activity. **c** Single-cell clones from the 3rd round sorted library were characterized by Tango/APJ β-arrestin assay. Stimulation by 100 nM apelin 13 and 0.1% DMSO were set as 100% activation and 0% activation, respectively. **d** IC$_{50}$ values of nine soluble sdAbs cloned from positive cell clones by PathHunter β-arrestin assay. IC$_{50}$ values and 95% confidence intervals are shown. **e** Correlation between the inhibitory activity of the cell clones and the IC$_{50}$s of the corresponding soluble sdAbs by β-arrestin assay. Inhibitory activity of the cell clones was done by Tango assay. Pearson correlation analysis was used in the association study. R square, *P* value and 95% confidence bands are shown.

recruitment reporter assay. This new method directly links antibody genotype with phenotype, allowing quick identification of functional antibodies. Compared to binding-based screening methods, such as hybridoma screening and phage display, this function-based high-throughput screening method decreases the screening scale and minimizes the workload for hits identification. The GPI-anchoring system may be advantageous over covalent linking the antibodies to a transmembrane domain in autocrine signaling-based cell sorting and screening[43,44]. The colocalization of GPI and GPCRs in lipid rafts facilitates GPI-anchored antibody interactions with the GPCR target[24,27]. Furthermore, soluble antibodies can be obtained by directly digesting GPI-anchored antibodies with PI-PLC for a quick confirmation of biofunction prior to antibody gene amplification and subcloning, which can further minimize the workload and speed up the process for the identification of functional antibodies. We used β-arrestin recruitment-based cell sorting and screening instead of monitoring G protein-based signaling. The advantage of β-arrestin recruitment-based cell sorting and screening is that it does not depend on knowledge of the G protein signaling specificity of the target receptor. β-arrestin recruitment upon receptor activation is a common pathway among GPCRs, so this new method can be used to identify antibody antagonists and, more importantly, antibody agonists to any GPCR targets.

We validated this method by using APJ as a model GPCR target and identified a panel of APJ sdAb antagonists with diverse epitopes and, for the first time, to our knowledge, an sdAb agonist from an immune sdAb library. We previously isolated a panel of potent APJ sdAb antagonists and non-functional binders from the same immune library by phage display, but we were not able to isolate any sdAb agonists from the library by phage display.

Phage display panning seems to enrich for high-affinity binders to the dominant epitopes of the antigen. Epitope localization studies suggest that phage display-derived sdAb antagonists and non-functional binders dominantly bind to the ECL2 and the N-terminus of APJ, respectively. APJ ECL2 is critical for ligand binding and plays an important role in mediating ligand-induced receptor activation. It is understandable that the binding of sdAbs to ECL2 blocks the binding of apelin 13 to the ligand binding pocket, resulting in competitive antagonist activity, while the binding of sdAbs to the N-terminus does not affect the ligand binding. Unlike binding-based panning and screening of phage-displayed antibody library, function-based cell sorting of GPI-anchored antibody library described in this study facilitated the isolation of target-specific antibody antagonists that bind to diverse epitopes and have a broad range of antagonist activity. Most likely, function-based sorting of single antibody-displayed cells does not generate a pressure on cells to compete for the receptor, unlike the panning of phage-displayed antibody library, in which binders compete for a limited amount of antigen proteins. More importantly, the potency of isolated sdAb antagonists correlates with the inhibitory activity of the corresponding cell clones in the Tango/APJ β-arrestin assay. Therefore, one can isolate antibody antagonists with desired (high, moderate, or low) potency based on the inhibitory activity of the cell clones. Antibody agonists to GPCRs are rare, but the new method described in this study allows us to successfully identify the potent sdAb agonist JN300 to human APJ.

We described here the cell surface display of camelid sdAb libraries by anchoring them to GPI. In this study, we also tested sdAb-Fc fusions for anchoring to GPI and found that GPI-anchored sdAb-Fcs display well on cell surface and retain their

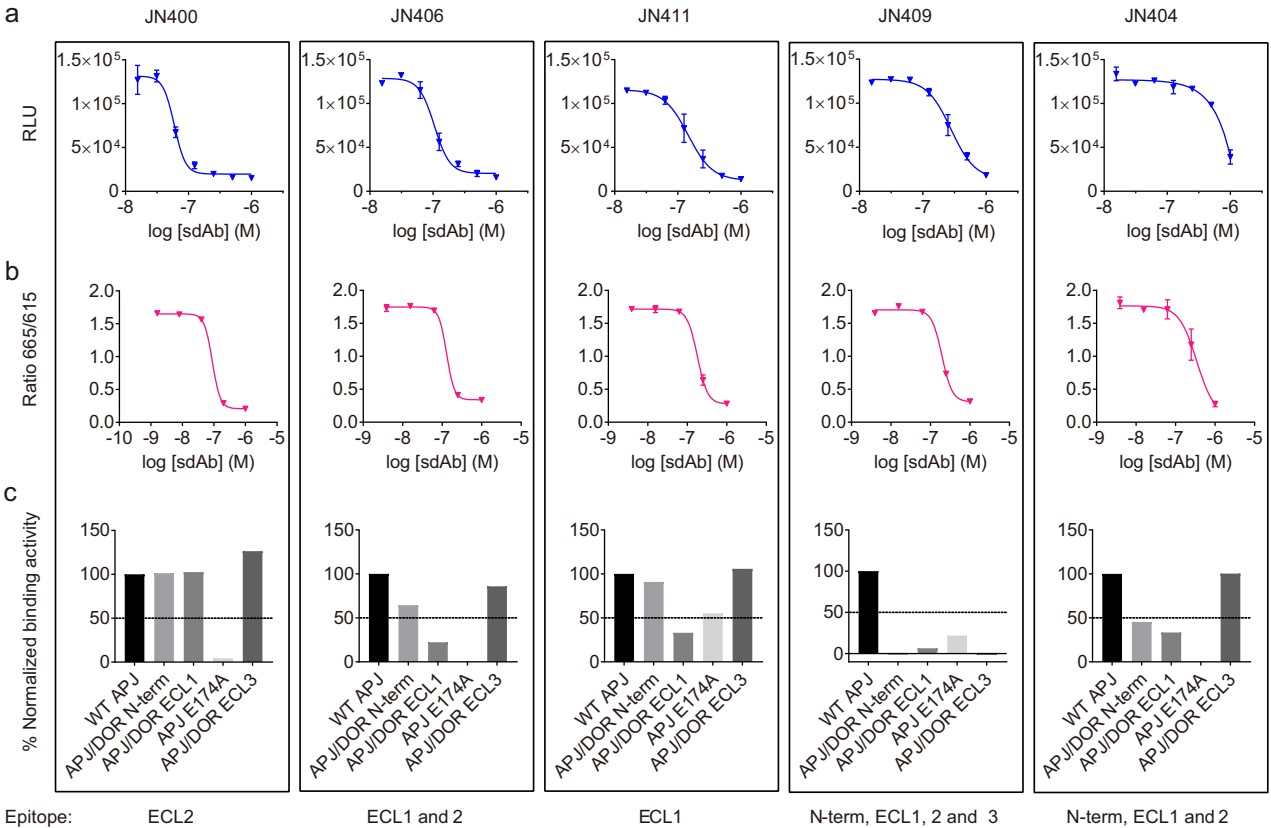

**Fig. 3 Characterization and epitope localization of isolated sdAb antagonists. a, b** Potency of isolated sdAb antagonists by PathHunter β-arrestin assay (**a**) and by cAMP assay (**b**). **c** Binding of the isolated sdAb antagonists to APJ/DOR domain-swapped mutants or APJ E174A site mutant relative to wild type (WT) APJ by flow cytometry. The 1st group includes JN400, JN402, JN413, and JN414 that bind to ECL2. Only JN400 is shown as a representative. The 2nd group include JN406 and JN412 that bind to both ECL1 and ECL2. Only JN406 is shown as a representative. The 3rd group include JN411 that binds to ECL1 only. The 4th group include JN401 and JN409 that are sensitive to the changes either in the N-terminus or in any of the ECLs. JN409 is shown as a representative. The 5th group include JN404 that is sensitive to the changes either in the N-terminus or in ECL1 or ECL2. APJ/DOR N-term: APJ and DOR N-terminus-swapped mutant; APJ/DOR ECL1: APJ and DOR ECL1-swapped mutant; APJ E174A: APJ E174 alanine substitution mutant; APJ/DOR ECL3: APJ and DOR ECL3-swapped mutant. Data are expressed as mean with SD in a, b.

biological functions, suggesting that Fab or IgG libraries could also be anchored to GPI for cell surface display and for function-based cell sorting and screening. To transfer a large phage-displayed antibody library to a lentiviral vector, library size might be a concern. We used one round of phage library panning prior to the transfer to enrich the binders without losing diversity. The introduction of the one round of binding-based phage library panning may lead to the possible loss of agonistic clones, which partially explains why we identified only one sdAb agonist in this campaign. In addition, we noticed that constitutive activation of the cells by GPI-anchored agonist antibodies led to the loss of agonist antibody genes in the cells or even cell apoptosis, which results in the failure in recovering the agonist sdAb genes following function-based cell sorting and screening. It is also possible that the cells were activated by agonist antibodies displayed by the neighbor cells when the cell density was high during sorting, which may lead to false positive signal. Inducible expression of GPI-anchored antibodies may alleviate this negative effect and enhance the sorting and screening efficiency in isolating agonist antibodies.

APJ functional antibodies identified in this study may have the potential for antibody therapeutics. To our knowledge, APJ sdAb agonist JN300 is the first antibody agonist identified so far against class A GPCRs, the largest and most important GPCR family for drug discovery. JN300 may have the therapeutic potential for the

treatment of chronic heart failure. Since APJ is also involved in angiogenesis and elevated APJ was observed in solid tumors[35,36], the APJ sdAb antagonists identified in this study may have the potential to treat solid tumors by blocking tumor angiogenesis. This function-based antibody high-throughput screening method can be applied to all GPCR targets. It can also be adapted to isolating functional antibodies against non-GPCRs by using reporter gene assays monitoring the downstream signaling pathways. We believe that the new method described here will greatly facilitate the identification of GPCR functional antibodies with therapeutic potentials, which may create a new paradigm for antibody therapeutics.

## Methods
**Cell lines and culture media.** Unless otherwise specified, all cell lines and culture reagents were from Invitrogen. HEK293FT cell was maintained in Dulbecco's modified Eagle's medium (DMEM) with 10% FBS, 100 U per ml penicillin, 100 μg per ml streptomycin, 0.1 mM non-essential amino acids (NEAA), 1 mM sodium pyruvate. HEK293FT/huAPJ cell line, HEK293FT/huAPJ domain-swapped mutants, and HEK293FT/huSSTR2 cell line were in-house generated by lipofectamine LTX transfection and maintained in HEK293FT cell culture medium containing 1.5 μg per ml puromycin. PathHunter CHO-k1/APJ β-arrestin cell line was obtained from DiscoverX and maintained in DMEM-F12 medium supplemented with 10% FBS, 100 U per ml penicillin, 100 μg per ml streptomycin, 500 μg per ml geneticin, and 200 μg per ml hygromycin B. CHO-k1/APJ β-arrestin cells with GPI-anchored sdAbs were in-house generated and maintained in PathHunter cell culture medium containing 5 μg per ml puromycin. U2OS AGTRL1 (APJ)

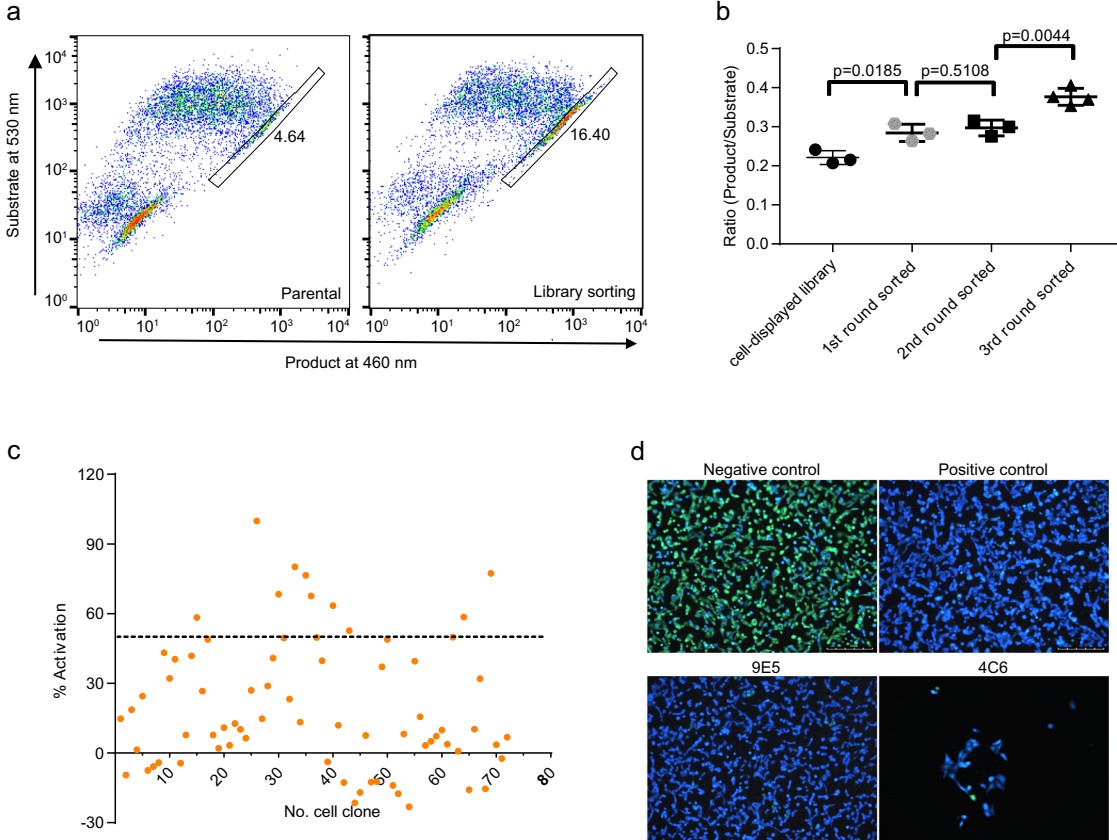

**Fig. 4 Function-based cell sorting and screening for agonistic cell clones. a** Dot-plot of Tango/APJ sdAb cell-displayed library following substrate loading. Cells with high ratio of product/substrate (460 nm/530 nm) were sorted out. The gating window and the percent of gated cell population in the 3rd round cell sorting are shown. **b** β-arrestin recruitment activity of the parental library and the 1st, 2nd and 3rd round sorted cell populations by Tango/APJ β-arrestin assay. Data are expressed as mean with SD. Statistical analysis was done using unpaired two-tailed t test. The higher product/substrate ratio, the higher agonist activity. **c** Percentage activation of monoclonal cells by Tango/APJ β-arrestin assay. Stimulation by 100 nM apelin 13 and 0.1% DMSO were set as 100% activation and 0% activation, respectively. **d** Confocal microscopy of single-cell clones tested for β-arrestin recruitment activity. Negative control: Parental Tango/APJ β-arrestin assay cells. Positive control: Parental Tango/APJ β-arrestin assay cells treated with 15 nM apelin 13. Two representative single-cell clones, 9E10 and 4C6, are shown. A scale bar of 250 μm is shown in each image.

stable cell line used in receptor internalization assay was purchased from DiscoverX (Cat. 93‒0710C3) and maintained in AssayComplete Cell Culture Kit 103 (DiscoverX) with 0.25 μg per ml puromycin, 250 μg per ml hygromycin B and 500 μg per ml geneticin. FreeStyle 293F cell line was maintained in FreeStyle 293 expression medium. Tango/APJ β-arrestin assay cell line was maintained in McCoy's 5A medium (modified) with 10% dialyzed FBS, 0.1 mM NEAA, 100 U per ml penicillin, 100 μg per ml streptomycin, 1 mM sodium pyruvate, 25 mM HEPES (pH 7.3), 200 μg per ml zeocin, 50 μg per ml hygromycin B, and 100 μg per ml geneticin. Tango/APJ β-arrestin assay cells with GPI-anchored sdAbs were in-house generated and maintained in Tango/APJ β-arrestin assay cell culture medium containing 0.5 μg per ml puromycin. All cells were maintained in 5% $CO_2$ incubators with a humidified atmosphere at 37 °C.

**Plasmids**. A pHEN2-based phagemid vector was used in the construction of camelid immune sdAb library. The third-generation lentiviral transfer vector pHBLV-puro, lentiviral packaging vector psPAX2 and envelope vector pMD2.G were purchased from Han bio, China. pHBLV-puro vector contains a CMV promoter upstream a multiple cloning site (MCS). We modified pHBLV-puro vector by inserting a signal peptide right after the CMV promoter, and a his6 tag, a $(G_4S)_5$ linker and a GPI attachment signal sequence, the C-terminal 34 amino acid residues of DAF, after the MCS. The modified lentiviral vector was designated pHBLV-puro-GPI. Recombinant sdAb or sdAb-Fc genes were synthesized by GenScript (Nanjing, China) and cloned into pHBLV-puro-GPI vector via Sph I and Nhe I sites.

**Generation of recombinant lentiviruses**. All lentivirus-related experiments were performed at BSL2 lab. Briefly, $5 \times 10^6$ HEK293FT cells were seeded in 10 ml of growth medium in a 10-cm cell culture dish. After cultivation for 18–24 h, cells were transfected with 5 μg of pHBLV-puro-GPI plasmid, 10 μg of psPAX2 plasmid, and 5 μg of pMD2.G using Lipofectamine 2000 (Invitrogen) as transfection reagent.

Following incubation for 16 h, culture supernatant was removed and replaced with 10 ml fresh growth medium containing 10 mM sodium butyrate (Sigma). Following further incubation for 8 h, the medium was changed to growth medium again, and cells were incubated for 24 h. The culture supernatants were then harvested and concentrated by using PEG-it Virus Precipitation Solution (System Biosciences) according to the manufacturer's instructions. The concentrated viruses were aliquoted and stored at −80 °C.

**Titration of recombinant lentiviruses**. The recombinant lentiviruses were titrated in HEK293FT cells. Briefly, $1 \times 10^5$ cells in 500 μl cell culture medium were seeded into each well in a 12-well plate. After overnight cultivation, 500 μl of 10-fold serial diluted recombinant lentiviruses were added to each well for virus transduction. After cultivated at 37 °C for 48–72 h, the expression level of the transduced gene was measured by flow cytometry. The virus transducing units (TU) were calculated according to the formula: TU per ml = cell number × positive cell population %/virus volume × dilution fold.

**Generation of stably transduced cells for POC study**. To generate stably transduced cell lines, $1 \times 10^5$ CHO-k1/APJ or $2 \times 10^5$ Tango/APJ β-arrestin cells per well were seeded onto a 12-well plate. After cultivation for 16 h, recombinant lentiviral viruses expressing sdAbs were used to infect cells at multiplicity of infection of 10, together with 1/200 volume of TransDux Max and 1/5 volume of Max Enhancer (System Biosciences) in the same well. The expression level of the transduced gene was measured by flow cytometry 48–72 h post infection. A minimum percentage of positive cells after transduction should reach 98%. In all, 5 μg per ml puromycin (CHO-k1/APJ cells) or 0.5 μg per ml puromycin (Tango/APJ cells) was added to generate stable cell lines.

**Flow cytometry**. To test the expression level of transduced genes, $2 \times 10^5$ lentivirus transduced cells were collected, washed twice with 1X PBS, and incubated with a

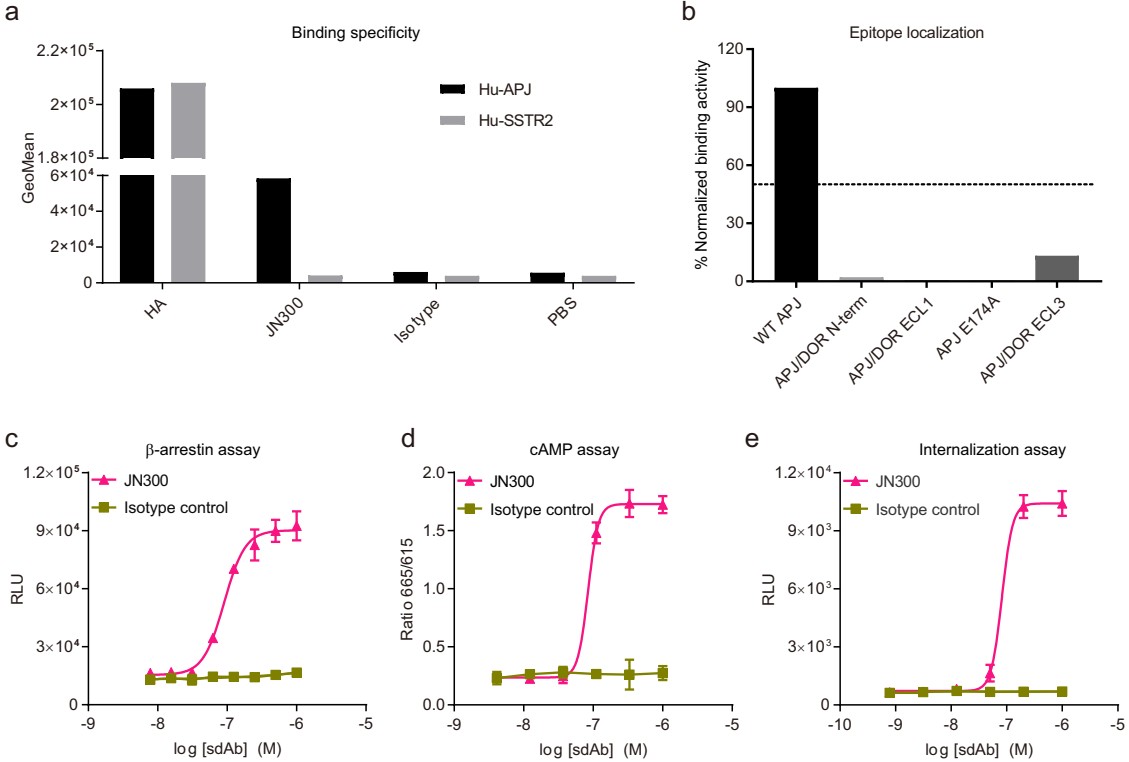

**Fig. 5 Characterization of sdAb agonist JN300 for binding and function. a** Flow cytometry of 293FT/HA-APJ and 293FT/HA-SSTR2 stable cell pools stained with JN300. **b** Binding of JN300 to WT APJ, APJ/DOR domain-swapped mutants, and APJ E174A mutant relative to its binding to WT APJ by flow cytometry. **c–e** Agonist activity of soluble sdAb JN300 in PathHunter β-arrestin assay (**c**), LANCE cAMP assay (**d**) and PathHunter-based receptor internalization assay (**e**). An irrelevant sdAb served as isotype control in (**a** and **c–e**). Data are expressed as mean with SD.

mouse anti-his antibody (1 μg per ml, GenScript, A00186) at 4 °C for 1 h. Cells were then washed twice with 1x PBS, and stained with Alexa Fluor 488 conjugated goat anti-mouse antibodies (2 μg per ml, Invitrogen, A-11029) at 4 °C for 30 min. After that, cells were washed twice with 1X PBS again and resuspended in 60 μl 1X PBS. FACS analysis was performed in an IntelliCyt iQue Screener (Sartorius). In the PI-PLC treatment experiment, $5 \times 10^5$ transduced CHO-k1/huAPJ cells were treated with 0.2 units of PI-PLC (Invitrogen, P-6466) or 1X PBS and rocked at 4 °C for 2 h. Cells were then stained with the same procedure.

For epitope localization of isolated APJ-specific sdAbs, $2 \times 10^5$ HEK293FT/huAPJ cells, or HEK293FT/huAPJ domain-swapped mutants, or HEK293FT/huSSTR2 cells were stained with 250 nM sdAbs at 4 °C for 1 h. The rest of the procedure were the same as described above. The receptor expression level on cell surface was determined by directly staining the cells with Alexa Fluor 488 conjugated anti-HA antibodies (5 μg per ml, Invitrogen, A-21287) at 4 °C for 30 min. The ratio of the GeoMean with sdAb staining relative to the GeoMean with anti-HA staining was calculated using the formula: Ratio = $(GeoMean_{sdAb} - GeoMean_{PBS})/(Geomean_{HA} - GeoMean_{PBS})$ except for ECL3 mutant. In the case of ECL3 mutant, the highest GeoMean of sdAb staining was much higher than the GeoMean of anti-HA antibody staining, so the ratio was calculated as the GeoMean with sdAb staining relative to the highest GeoMean with sdAb staining using the formula: Ratio of ECL3 = $(GeoMean_{sdAb} - GeoMean_{PBS})/(GeoMean_{highest\ sdAb} - GeoMean_{PBS})$. All the ratios were normalized to the binding to WT APJ receptor by the formula: % Normalized binding activity = $Ratio_{receptor}/Ratio_{WT\ APJ\ receptor} \times 100\%$.

**Construction of GPI-anchored sdAb library.** A bactrian camel (*Camelus bactrianus*) was immunized with human APJ nanodiscs by subcutaneous injection for five times (primary immunization and four boosts) with 3 weeks interval using Freund complete and incomplete adjuvants. IACUC guidelines were followed with animal subjects. The PBMCs from the final bleed of the immunized camel were used to prepare total RNAs for the amplification of the VHH genes by RT-PCR and nest PCR following the protocol previously described. Briefly, a pair of primers that anneal to heavy chain antibody leader sequence (pLead: 5′-GTCCTGGCTGCT CTTCTACAAGG-3′) and CH2 (pCH2: 5′-GGTACGTGCTGTTGAACTGT TCC-3′) were used for the 1st PCR. A second pair of primers that anneal to VHH FR1 (pFR1: 5′-CAGCCGGCCATGGCCSAKGTGCAGCTGGTGGAGTCTGG-3′) and FR4 (pFR4: 5′-ATGATGATGTGCGGCCGCTGAGGAGACRGTGACC WG-3′) were used for the nest PCR. The PCR products were digested with BssH II and Nhe I, gel-extracted and cloned to a modified pHEN2 phagemid vector, resulting in a large immune sdAb library containing $2 \times 10^9$ individual clones. APJ

immune phage-displayed sdAb library was subject to one round panning against 293FT/huAPJ cells to enrich the binders prior to transferring the immune repertoire to the lentiviral vector. Briefly, $1 \times 10^{13}$ phage particles from the phage library was depleted with $5 \times 10^7$ 293FT/huSSTR2 cells, and then panned against $1 \times 10^7$ 293FT/huAPJ cells in solution. Unbound phage particles were removed by washing with PBST for 10 times and with PBS for another 10 times. Bound phage was eluted with 600 μl 100 mM Triethanolamine (Sigma-Aldrich) by incubation at room temperature for 10 min followed by neutralization with 300 μl 1 M Tris-HCl (pH 8.0). Eluted phage were then added to 10 volumes of log-phase TG1 cells ($OD_{600} = 0.5$ to 1.0) in 2YT medium and incubated at 37 °C for 30 min. In all, 50 μl phage-infected TG1 cells were used to make 10-fold serial dilutions in 2YT medium and plated each serial dilution onto 2YTCG (50 μg per ml Carbenicillin, 2% glucose) plates for phage titration. The output of phage library was $4 \times 10^6$. All the other phage-infected TG1 cells were plated onto a Nunc square bioassay dish and incubated at 30 °C overnight. Bacteria were scraped off from the dish and amplified for phagemid preparation. At this step, the bacterial inoculum should be at least 10-fold of the phage output and the $OD_{600}$ value of the amplified bacteria should be <3.0. A total of 875 ng of digested and gel-extracted inserts were ligated with 5 μg of digested and purified lentiviral vector pHBLV-puro-GPI using *Sph* I and *Nhe* I sites. The ligation product was used to transform electrocompetent *E. coli.* strain Stbl3 (Huayueyang Bio, Beijing, China), resulting in a combinatorial lentiviral sdAb library with a size of $4 \times 10^7$ individual clones. The combinatorial lentiviral sdAb library was packaged into lentiviruses as described above. A total of $3.5 \times 10^8$ Tango/APJ β-arrestin assay cells from 35 dishes (15-cm dish, $1 \times 10^7$ cells per dish) were then used in the transduction with recombinant lentiviruses at multiplicity of infection of 0.3. The GPI-anchored sdAb cell-displayed library size was defined as the number of his 6 tag-positive cells measured by flow cytometry 48 h post infection. In this practice, his 6 tag-positive cells were on average 27% in each dish, so the actual library size in mammalian cells was $9.45 \times 10^7$. The non-transduced cells were eliminated by the treatment with 0.5 μg per ml puromycin.

**cAMP assay.** Cellular cAMP was measured by Lance Ultra cAMP kit (Perkin Elmer) according to manufacturer's instruction. Briefly, 2000 cells in 10 μl assay buffer (HBSS + 5 mM HEPES + 0.1% BSA) were added into each well on a 96-half well assay plate and stimulated with test antibodies (at agonist mode), or with apelin 13 ligand at $EC_{80}$ dose together with test antibodies (at antagonist mode) in the presence of 2.5 μM forskolin. The plates were incubated at 37 °C for 30 min followed by the addition of 10 μl of 4X Eu-cAMP and 10 μl of 4X Ulight-Anti-cAMP working solutions. The mixture was incubated at room temperature for 60 min and then read

with Envision plate reader using a recommended setting (Excitation at 320 nm and Emission at 615 nm and 665 nm). The cellular cAMP level was expressed as the signal ratio of $Em_{665 nm}$ to $Em_{615 nm}$. In all, 100 nM apelin 13 was used as a positive control and 1% DMSO as a negative control on each plate.

**PathHunter β-arrestin recruitment assay.** The PathHunter β-arrestin recruitment assay was performed according to manufacturer's instructions with minor modifications. Briefly, cells were seeded on 384-well plates (Corning, #3707) at 5000 cells per well in Opti-MEM with 0.5% FBS and cultivated overnight. On the next day, test antibodies (at agonist mode) or apelin 13 ligand at $EC_{80}$ dose together with test antibodies (at antagonist mode) were added into the wells. After incubation at 37 °C for 1.5 h, detection reagent was added to each well. The mixture was incubated at room temperature for 60 min before reading with a luminance plate reader (Envision).

**PathHunter-based GPCR internalization assay.** The PathHunter-based GPCR internalization assay was performed according to manufacturer's instructions with minor modifications. Briefly, U2OS cells stably expressing human APJ were seeded on 384-well plates (Corning, #3707) at 5000 cells per well in Opti-MEM with 0.5% FBS and incubated for 24 h. On the second day, serially diluted antibodies were added into each well. After incubation at 37 °C for 3 h, detection reagent was added to each well. The mixture was incubated at room temperature for 60 min before reading with a luminance Plate reader (Envision or CLARIOstar).

**Tango assay with GPI-anchored sdAbs.** Briefly, parental cells or sdAb-anchored cells were seeded on 96-well plates (costar, #3603) at 58,000 cells per well in 40 μl FreeStyle 293 expression medium and cultivated for 48 h. On the third day, 10 μl FreeStyle 293 expression medium (at agonist mode) or 10 μl FreeStyle 293 expression medium containing apelin 13 at $EC_{80}$ dose (at antagonist mode) were added to the plates. Parental cells treated with 100 nM apelin 13 or 0.1% DMSO were used as positive or negative controls, respectively. After incubation at 37 °C for 5 h, 10 μl of LIVEBLAZER FRET B/G loading solution was added to the wells. The mixture was incubated at room temperature for 2 h before reading with a fluorescence plate reader (CLARIOstar, Excitation at 405 nm, Emission at 460 nm and 530 nm) or analyzed by flow cytometry. Cell-free control wells were used for background subtraction. The product/substrate ratio of cell clones was calculated as the signal at 460 nm to the signal at 530 nm using the formula: Ratio (product/substrate) = $(Em_{460sample} − Em_{460medium})/(Em_{530sample} − Em_{530medium})$. Stimulation by 100 nM apelin 13 and 0.1% DMSO were set as 100% activation and 0% activation, respectively. Percentage inhibition was calculated using the formula: % inhibition = $100\% − (Ratio_{sample} − Ratio_{DMSO})/(Ratio_{apelin13} − Ratio_{DMSO}) \times 100\%$. Percentage activation was calculated using the formula: % activation = $(Ratio_{sample} − Ratio_{DMSO})/(Ratio_{apelin13} − Ratio_{DMSO}) \times 100\%$.

**Function-based cell sorting and screening.** The GPI-anchored sdAb library in FreeStyle 293 expression medium was plated onto four to five 15-cm cell culture dishes and cultivated for 48 h in 5% $CO_2$ incubator at 37 °C for cell starvation. Total his-positive cell number should lager than the 10-fold size of panned-phage library.

For cell sorting at agonist mode, one-fifth volume of substrate solution were added to the cells on the third day. The cells were then incubated at room temperature for 2 h, followed by digestion with 0.05% Trypsin-EDTA solution (Invitrogen) and then washed three times with 1X PBS. Resuspended cells in 1X PBS supplemented with 2% BSA were sorted for the cells with high signal ratio of $Em_{460 nm}$ to $Em_{530 nm}$ using FACSJazz Cell Sorter (BD Biosciences).

For cell sorting at antagonist mode, on the third day, culture medium was replaced with fresh FreeStyle 293 expression medium containing apelin 13 at $EC_{80}$ dose and cultivated at 37 °C for 12–16 h. On the fourth day, one-fifth volume of substrate solution were added to the plates. The mixture was incubated at RT for 2 h for substrate loading. After cell digestion and washing, cells with low signal ratio of $Em_{460nm}/Em_{530nm}$ were sorted out.

Following the 3rd round sorting, single cell clones were seeded to 96-well assay plates (Corning, #3603). When the single cells grow to over 50% confluent, the culture medium was replaced with FreeStyle 293 expression medium and cells were incubated for 48 h at 37 °C. The Tango assay was then performed as described above. Positive clones were double checked under a fluorescent microscope (LEICA DMi8, Excitation at 409 nm). Antibody genes were recovered by one-step RT-PCR using a forward primer (5′-CTGCTGGGAATGCTGGTGG-3′) and a reverse primer (5′-GGTGGTGATGCAGGTCCTC-3′) and cloned into a TA cloning vector for sequencing.

**Antibody expression and purification.** Soluble sdAbs were expressed in FreeStyle 293 transient expression system (Invitrogen) using pTT5 vector. Briefly, $3 \times 10^7$ FreeStyle 293 F cells were transfected with 30 μg of pTT5 plasmid by poly-ethylenimine. After cultivation at 37 °C for 3–6 days, the culture supernatants were harvested for sdAb purification. His-tagged soluble sdAbs were purified by immobilized metal affinity chromatography (IMAC) as follow: 10 mM imidazole and 300 mM NaCl (final concentration) was added to the supernatant. The supernatant was then mixed with pre-washed Ni-NTA resin (Qiagen) (0.2 ml resin per 30 ml culture) and incubated at 4 °C for 1 h. The resin was washed with 5 column volume (CV) of 1X PBS containing 20 mM imidazole. Bound sdAbs were eluted by 3 CV of elution buffer (1X PBS supplemented with 300 mM imidazole) and the eluates dialyzed against 1X PBS to remove imidazole. Antibody concentration was measured by Nanodrop (E = 28, MW = 15.5) or BCA assay (Pierce™ BCA Protein Assay Reagent, microplate mode).

**Statistics and reproducibility.** Each sample was tested in duplicate or triplicate in functional assays and each functional assay was repeated twice using the same samples. Representative data were shown. Statistical analysis was done using GraphPad Prism 7 software. Unpaired two-tailed $t$ test was used to determine the statistical differences between different samples. Pearson correlation analysis (linear regression) was used to examine the association between experimental variables. $P$ value below 0.05 is considered as statistically significant for either test.

**Reporting summary.** Further information on research design is available in the Nature Research Reporting Summary linked to this article.

# Data availability

All data needed to evaluate the conclusions in the paper are present in the paper. Additional data and research materials related to this paper are available upon reasonable request by contacting the corresponding authors of this paper.

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

## Acknowledgements

We thank M. Zhang for supporting this study and for insightful discussions, X. Ma and Y. Song for helpful discussions, and I. Chen and M. Michaels for critical reading of the manuscript. The study was funded by Amgen Inc. and partially by a grant from Shanghai Pudong New Area Technology Development Postdoctoral Fund. H.R. was a postdoc fellow at the time the research was conducted.

## Author contributions

H.R. performed all experiments, analyzed the data, and drafted the manuscript. J.L. did preliminary test with Tango/APJ β-arrestin assay cells and provided important suggestions. N.Z. generated stable cell pools of APJ and DOR domain-swapped mutants for epitope localization. L.A.H. and Y.M. assisted in data interpretation and manuscript preparation. P.T. supported this study and provided valuable suggestions. J.X. co-supervised the study, assisted in data interpretation and manuscript preparation. M.-Y.Z. conceived and supervised the project, designed experiments, reviewed data, and wrote the paper.

## Competing interests
H.R., J.L., N.Z., L.A.H., Y.M., P.T., and M.-Y.Z. are employees of Amgen.
