## [Peer Review File · Communications Biology]

Reviewers' comments:

Reviewer #1 (Remarks to the Author):

The manuscript by Ren et al, highlights the development of a novel biofunction-based-panning process mainly for G-protein-coupled proteins. The ability of integrating the previous method by Zhang et al to identify agonistic and antagonistic antibodies against GPCR targets is very interesting and timely. The manuscript is well written albeit minor grammatical mistakes and citations. Enclosed are several suggestions to help improve the manuscript for consideration for publication in Communications Biology.

Minor comments:

In line 43 and 44, the invention of DNA technology for gene cloning etc, there is no reference provided.

Reference 3 used for the statement from line 45 to 49, please confirm if the reference is suitable.

Line 55: Antibody early discovery? Does the authors mean rapid discovery of antibodies instead?

Statement from line 55 to 58 should have reference

Line 59 to 60, the size of ribosome libraries does not have any reference.

Line 60 to 61: hybridoma technology and display technology: statement can be further clarified to include that hybridoma technologies not only provides natural pairing but also affinity matured sequences for antibodies. Whereas display technologies in a naïve library does not. Even so, the author should note that for immune libraries the situation is much like hybridoma technologies in terms of affinity maturation.

Line 95: May induce,... I suspect the sentence is hanging.

Line 129: Reference for the G-protein pathways.

Line 137, the immunized camels library should have a reference.

Line 182: is it report cell line or reporter cell line?

Line 263: it should be percentage not percent

There are some parts of the discussion which should have referneces. This is mainly to those parts discussing the mechanism of action.

Line 382: 37C please include the symbols.

Line 422: The percent of positive cells was usually over 98% after one round of transduction. This phrase should be rephrased as this could be arbitrary. It would be best to present it as a minimum percentage of positive cells after transduction should reach ??%.

Line 471 to 473: The amount of phagemids used for transformation was not defined.

Line 535: temperature symbol is missing

Figures: The figures with microscopy images should have the scales included.

Major comments:

The approach required the utilization of one round of panning: The eluted phage was determined as 4×10^6 . This phagemid was then cloned into the lentivirus to yield 4×10^7 size. Pls confirm if this was done in bacterial cell or mammalian cells? So what is the actual size in mammalian cells? Also, the method or process to determine the library size was not mentioned.

Is there any other measure of the single round panning?

The authors mentioned that 10 clones were obtained and characterized. It would be good to include the sequences of the 10 clones. Or at least the sequence analysis in terms of CDR length or V-gene family if the full identity of the clones has to be protected.

I would also suggest that the expression and purification profiles of the scDABs be included in supplementary.

For the epitope localization, it would be good for the readers that an actual model highlighting the epitopes be provided.

It would be good to have a sequence comparison analysis between the agonist and antagonist clones.

The authors mentioned in the discussion, the use of sDAB-Fc fusions. There was no data or reference provided for this data.

As the authors only identified one clone JN300 to show agonistic function, the authors should also consider discussing about the possible loss of clones from the introduction of the 1 round of panning. It is likely as it could support the numerous failures to obtain an agonistic clone using phage display panning.

Reviewer #2 (Remarks to the Author):

This is an overall well written manuscript with good quality data and figures.

GPCRs are one of the most attractive therapeutic target classes. Nevertheless, raising functional antibody against GPCRs remains challenging. The authors develop a function-based screening method for identifying antibody antagonists and agonists against GPCRs by combining mammalian cell surface display with the autocrine based screening technology. The author identified antagonists of apelin receptor with broader epitopes than those antibody from biopanning of the same calmid immunized antibody library and they also identified an antibody agonist which was not found by phage display. Here are my comments in detail.

1. I suggest that a schematic of Tango assay is included in Fig 1 to better demonstrate the method. The display level of antibody in library can be included in SI.

2. It is better to test JN300 in 1-2 other GPCR tango reporter cell lines to show the selectivity of the antibody.

3. The information of immunization and construction of phage display library should be briefly described in method and material.

4. There are some false positive hits when the agonist antibody was screened. Please discuss the possible reasons, e.g. the cells were activated by the agonist antibody displayed by the neighbour cells when the cell density was high.

Reviewer #3 (Remarks to the Author):

The authors have devised a method to identify antibody agonists and antagonists to GPCRs through phage display and beta-arrestin recruitment reporter assay. The design of the study was good. The authors have performed the experiments in a logical way. A careful English editing is required.

1. The introduction looks too lengthy and boring with historical events. The authors should concise the introduction that focuses on their method alone. This will help readers better understand the work.

2. Phage display is now one of the most widely used technology for antibody discovery and engineering. Why APJ was used as a model GPCR target among others? The authors should include the domain structure of APJ in the figure.

3. Map of all the plasmids used in this study should be given as supplementary material.

4. In the discussion section, the authors should briefly describe why GPCRs (transmembrane proteins) are difficult to generate antibodies against them.

5. In each of the figure legends, the authors should clearly mention the statistical analysis that they have carried out and in many cases significance is not mentioned.

Point-by-point responses to the editorial and reviewers' comments:

Reviewer #1 (Remarks to the Author):

The manuscript by Ren et al, highlights the development of a novel biofunction-based-panning process mainly for G-protein-coupled proteins. The ability of integrating the previous method by Zhang et al to identify agonistic and antagonistic antibodies against GPCR targets is very interesting and timely. The manuscript is well written albeit minor grammatical mistakes and citations. Enclosed are several suggestions to help improve the manuscript for consideration for publication in Communications Biology.

A: We thank Reviewer #1 for very positive comments on the manuscript. Below are the responses to the reviewer's minor and major comments.

Minor comments:

In line 43 and 44, the invention of DNA technology for gene cloning etc, there is no reference provided.

A: Four references for DNA technology are added.

Reference 3 used for the statement from line 45 to 49, please confirm if the reference is suitable.

A: It is replaced with a new reference now.

Line 55: Antibody early discovery? Does the authors mean rapid discovery of antibodies instead?

A: We changed the text to "rapid discovery of antibodies" as the reviewer suggested.

Statement from line 55 to 58 should have reference

A: One reference is added.

Line 59 to 60, the size of ribosome libraries does not have any reference.

A: One reference is added.

Line 60 to 61: hybridoma technology and display technology: statement can be further clarified to include that hybridoma technologies not only provides natural pairing but also affinity matured sequences for antibodies. Whereas display technologies in a naïve library does not. Even so, the author should note that for immune libraries the situation is much like hybridoma technologies in terms of affinity maturation.

A: We acknowledge that hybridoma technology provides affinity matured antibody sequences, but the introduction seems lengthy as indicated by reviewer 3, so we do not add more comments on the comparison of hybridoma technology with display technologies.

Line 95: May induce,... I suspect the sentence is hanging.

A: The half sentence is now removed. We apologize for the ignorance.

Line 129: Reference for the G-protein pathways.

A: One reference for G-protein pathways is added.

Line 137, the immunized camels library should have a reference.

A: It is our previous work and the manuscript is accepted in Science Advances. We may be able to refer to the paper before this manuscript is published.

Line 182: is it report cell line or reporter cell line?

A: It is reporter cell line.

Line 263: it should be percentage not percent

A: "Percent" is changed to "Percentage" throughout the manuscript.

There are some parts of the discussion which should have references. This is mainly to those parts discussing the mechanism of action.

A: Two reference papers are added to the discussion.

Line 382: 37C please include the symbols.

A: It is corrected now.

Line 422: The percent of positive cells was usually over 98% after one round of transduction. This phrase should be rephrased as this could be arbitrary. It would be best to present it as a minimum percentage of positive cells after transduction should reach ??%.

A: It is rephrased as follow: A minimum percentage of positive cells after transduction should reach 98%.

Line 471 to 473: The amount of phagemids used for transformation was not defined. The inserts were then batch-transferred from the phagemids to the lentiviral vector pHBLV-puro-GPI by digestion and ligation using Sph I and Nhe I sites.

A: how much inserts were used in ligation with lentiviral vector and how much ligation products were used in transformation of bacteria (which strain?)

Line 535: temperature symbol is missing

A: Temperature symbol is added now.

Figures: The figures with microscopy images should have the scales included.

A: The scales are added to Fig. 4.

Major comments:

The approach required the utilization of one round of panning: The eluted phage was determined as 4×10^6 . This phagemid was then cloned into the lentivirus to yield 4×10^7 size. Pls confirm if this was done in bacterial cell or mammalian cells? So what is the actual size in mammalian cells? Also, the method or process to determine the library size was not mentioned.

A: The titration of eluted phage library was done in bacterial cells (*E. coli*, strain TG1). The titration of recombinant lentivirus was done in mammalian cells (Tango/APJ β -arrestin assay cells). The method to determine the lentivirus-transduced antibody cell-displayed library is described in the revised “Methods” as follow: The library size was defined as the number of his 6 tag-positive cells measured by flow cytometry 48 h post infection.

Is there any other measure of the single round panning?

A: There is no other measure of the single round panning against target-expressing cells. It is a regular one round cell panning.

The authors mentioned that 10 clones were obtained and characterized. It would be good to include the sequences of the 10 clones. Or at least the sequence analysis in terms of CDR length or V-gene family if the full identity of the clones has to be protected.

A: We apologize that we cannot release the sequences of the 10 clones, but we added a supplementary Table (Table S1) to summarize the sequence analysis result of the 10 clones including the V-gene usage and CDR lengths.

I would also suggest that the expression and purification profiles of the scDABs be included in supplementary.

A: The expression profile of the isolated 10 sdAb antagonists and one agonist is shown in Table S1.

For the epitope localization, it would be good for the readers that an actual model highlighting the epitopes be provided.

A: We added a schematic of APJ/DOR domain-swapped mutants highlighting the changed portions (Fig. S3) in the revised manuscript.

It would be good to have a sequence comparison analysis between the agonist and antagonist clones.

A: As shown in the newly added Table S1, sdAb agonist JN300 used one of the most common V-gene families in camelid V-gene repertoires and its CDR3 is also on average in length. But we do notice that the CDR1 of JN300 is short compared to the 10 antagonistic clones. We are not sure if this shortness is related to the agonistic activity. Since we cannot release the sequences of the isolated functional sdAbs at this step, we cannot describe more about the sequence comparison. We are currently trying to co-crystallize APJ in complex with JN300. Hope it will tell us more about the differences between the antagonistic and agonistic sdAbs.

The authors mentioned in the discussion, the use of sDAB-Fc fusions. There was no data or reference provided for this data.

A: A small figure is added to Fig. S1 showing the GPI-anchoring of Fc fusions.

As the authors only identified one clone JN300 to show agonistic function, the authors should also consider discussing about the possible loss of clones from the introduction of the 1 round of panning. It is likely as it could support the numerous failures to obtain an agonistic clone using phage display panning.

A: We agree with the reviewer that one round of phage library panning may lead to the possible loss of agonistic sdAbs. We added one sentence in the Discussion to acknowledge this limitation of the method described in this paper.

Reviewer #2 (Remarks to the Author):

This is an overall well written manuscript with good quality data and figures. GPCRs are one of the most attractive therapeutic target classes. Nevertheless, raising functional antibody against GPCRs remains challenging. The authors develop a function-based screening method for identifying antibody antagonists and agonists against GPCRs by combining mammalian cell surface display with the autocrine based screening technology. The author identified antagonists of apelin receptor with broader epitopes than those antibody from biopanning of the same camelid immunized antibody library and they also identified an antibody agonist which was not found by phage display.

Here are my comments in detail.

1. I suggest that a schematic of Tango assay is included in Fig 1 to better demonstrate the method. The display level of antibody in library can be included in SI.

A: A schematic of Tango assay is added to Fig. S2 to illustrate the method. The display level of sdAbs in the library is also included in Fig. S2, as well as the schematic of cell-displayed library construction.

2. It is better to test JN300 in 1-2 other GPCR tango reporter cell lines to show the selectivity of the antibody.

A: We indeed tested JN300 in another GPCR (SSTR2) Tango reporter cell line along with the APJ Tango cell line. The binding specificity of JN300 was confirmed as shown in the revised Fig. 5.

3. The information of immunization and construction of phage display library should be briefly described in method and material.

A: We provided more details in camel immunization and phage library construction in the revised “Methods”, including the immunogen format, immunization protocol, phagemid name and method for library construction. Please check the details in the revised manuscript.

4. There are some false positive hits when the agonist antibody was screened. Please discuss the possible reasons, e.g. the cells were activated by the agonist antibody displayed by the neighbour cells when the cell density was high.

A: We noticed that there are some false positive cell hits when we screened for sdAb agonists. There are two possible reasons: 1. Constitutive activation by agonist

antibodies may lead to the loss of antigen genes or even cell death, resulting in the failure in agonist sdAb gene recovery. 2. It is possible that the cells were activated by agonist antibodies displayed by the neighbor cells when the cell density was high during sorting. We discussed the reasons in the revised manuscript.

Reviewer #3 (Remarks to the Author):

The authors have devised a method to identify antibody agonists and antagonists to GPCRs through phage display and beta-arrestin recruitment reporter assay. The design of the study was good. The authors have performed the experiments in a logical way.

A careful English editing is required.

A: We thank Reviewer # s for the positive comments! This manuscript was edited by two native English speakers.

1. The introduction looks too lengthy and boring with historical events. The authors should concise the introduction that focuses on their method alone. This will help readers better understand the work.

A: We have shortened the “Introduction” by removing some historical events.

2. Phage display is now one of the most widely used technology for antibody discovery and engineering. Why APJ was used as a model GPCR target among others? The authors should include the domain structure of APJ in the figure.

A: We corrected the grammatic error in the sentence. Phage display is now one of the most widely used technologies for antibody discovery and engineering.

APJ was used as a model GPCR target in this study for two reasons: 1). APJ is a class A GPCR. Compared to class B GPCRs, class A GPCRs pose more challenges for generating target-specific binders and functional antibodies due to the short N-terminus of class A GPCRs. 2). For comparison of binding-based phage display library panning and screening with function-based GPI-anchored cell library sorting and screening. We tried the conventional route (animal immunization and phage display) to identify APJ agonist antibodies without success. Using APJ as a model target in developing an HTS method for functional antibodies makes sense for method comparison.

We added a schematic to show APJ domain structure (see Fig. S3)

3. Map of all the plasmids used in this study should be given as supplementary material.

A: Map of lentiviral vector pHBLV-puro-GPI is provided in the revised manuscript.

4. In the discussion section, the authors should briefly describe why GPCRs (transmembrane proteins) are difficult to generate antibodies against them.

A: We discussed the challenges for generating GPCR functional antibodies in the “Introduction”. We now added a few sentences in the Discussion to emphasize the

challenges for generating functional antibodies against class A GPCRs. APJ is a class A GPCR.

5. In each of the figure legends, the authors should clearly mention the statistically analysis that they have carried out and in many cases significance is not mentioned.

A: We added statistical analysis results to Fig. 2 and 4, as well as Fig. S1.

REVIEWERS' COMMENTS:

Reviewer #1 (Remarks to the Author):

The authors have addressed all my concerns.

Reviewer #2 (Remarks to the Author):

The authors answered all my questions. I believe it is acceptable for publication in Communications Biology.